# Long-Term Survival and Clinicopathological Implications of DNA Mismatch Repair Status in Endometrioid Endometrial Cancers in Hong Kong Chinese Women

**DOI:** 10.3390/biomedicines9101385

**Published:** 2021-10-04

**Authors:** Jacqueline Ho Sze Lee, Joshua Jing Xi Li, Chit Chow, Ronald Cheong Kin Chan, Johnny Sheung Him Kwan, Tat San Lau, Ka Fai To, So Fan Yim, Suet Ying Yeung, Joseph Kwong

**Affiliations:** 1Department of Obstetrics and Gynaecology, Faculty of Medicine, The Chinese University of Hong Kong, Hong Kong 999077, China; lautatsan@cuhk.edu.hk (T.S.L.); sfyim@cuhk.edu.hk (S.F.Y.); carolyeung@cuhk.edu.hk (S.Y.Y.); j.kwong@keele.ac.uk (J.K.); 2Department of Anatomical and Cellular Pathology, Faculty of Medicine, The Chinese University of Hong Kong, Hong Kong 999077, China; joshuali@cuhk.edu.hk (J.J.X.L.); chit@cuhk.edu.hk (C.C.); ronaldckchan@cuhk.edu.hk (R.C.K.C.); shkwan@cuhk.edu.hk (J.S.H.K.); kfto@cuhk.edu.hk (K.F.T.); 3School of Medicine, Faculty of Medicine and Health Sciences, Keele University, Newcastle-under-Lyme ST5 5BG, UK

**Keywords:** endometrioid endometrial cancer, DNA mismatch repair (MMR), MMR deficient (dMMR), long-term survival

## Abstract

To investigate the role of DNA mismatch repair status (MMR) in survival of endometrioid endometrial cancer in Hong Kong Chinese women and its correlation to clinical prognostic factors, 238 patients with endometrioid endometrial cancer were included. Tumor MMR status was evaluated by immunohistochemistry. Clinical characteristics and survival were determined. Association of MMR with survival and clinicopathological parameters were assessed. MMR deficiency (dMMR) was found in 43 cases (16.5%). dMMR was associated with poor prognostic factors including older age, higher stage, higher grade, larger tumor size and more radiotherapy usage. Long-term survival was worse in dMMR compared to the MMR proficient group. The dMMR group had more deaths, shorter disease-specific survival (DSS), shorter disease-free survival (DFS), less 10-year DSS, less 10-year DFS, and more recurrence. The 5-year DSS and 5-year DFS in the dMMR group only showed a trend of worse survival but did not reach statistical significance. In conclusion, dMMR is present in a significant number of endometrioid endometrial cancers patients and is associated with poorer clinicopathological factors and survival parameters in the long run. dMMR should be considered in the risk stratification of endometrial cancer to guide adjuvant therapy and individualisation for longer follow up plan.

## 1. Introduction

Endometrial cancer is the most common gynaecological cancer in the developed world and its incidence is on a continual rise. Endometrial carcinogenesis is driven by defects in signal transduction pathways such as the DNA mismatch repair (MMR) pathway, p53 pathway [1], phosphatidylinositol 3 kinase (PI3K)–AKT pathway, and WNT/ β-catenin signalling pathway [2]. Microsatellite instability (MSI) and phosphatase and tensin homolog (*PTEN*) mutation are the commonest genetic alterations in endometrial cancer [3].

The MMR system is a strand-specific DNA repair system which maintains genomic stability. MMR proteins, including MutL Homolog 1 (MLH1), MutS Homolog 2 (MSH2), MutS Homolog 6 (MSH6) and PMS1 Homolog 2, Mismatch Repair System Component (PMS2), are responsible for repairing base-base mismatch during DNA replication. MMR system also promotes cell cycle arrest and programmed cell deaths in response to DNA damage [4]. Microsatellites are short repetitive nucleotide sequences of DNA, which are prone to slippage and replication errors due to its repetitive structure. Therefore, when the MMR system is inactivated, it could lead to accumulation of DNA replication errors and MSI (in which the number of repeated DNA bases in a microsatellite is different from what it was when the microsatellite was inherited. In addition, inactivation of MMR system increases spontaneous mutations in the cells and is associated with cancers [4].

Deficiencies in MMR proteins were reported in 20% to 30% of endometrial cancers [5,6]. MMR deficiency can be somatic or germline. The majority of deficient mismatch repair deficiency in endometrial cancers is sporadic resulting from somatic mutations [3]. The presence of a germline mutation is suggestive of Lynch syndrome, a hereditary cancer syndrome associated with multiple cancers including colorectal and endometrial cancer [7]. Lynch syndrome is caused by an autosomal dominant germline mutation of one of the MMR genes, *MLH1*, *MSH2*, *MSH6* and *PMS2* [7]. 

There are three ways to detect defects in MMR system: immunohistochemistry (IHC), polymerase chain reaction (PCR)-based assays and next-generation sequencing (NGS) [8]. IHC can identify defects in MMR protein expression, classifying tumors as MMR proficient (pMMR) or MMR deficient (dMMR) [9,10]. PCR and NGS can evaluate for MSI, classifying tumors into high (MSI-H) or low (MSI-L) or stable levels (MSS) of MSI [8,9,10,11]. A study found a high correlation of IHC and PCR, with a concordance rate of 96% [9], while another study showed that IHC alone missed 17% of MSI cases identified by PCR [12]. The 2020 ESGO/ESTRO/ ESP guideline on endometrial cancer had recommended IHC as the preferred approach to identify defects in MMR gene because of its wide availability and cost-effectiveness [13].

Endometrial cancer patients generally have good prognosis with a 5-year survival rate of around 80% [14]. However, some cases (~20%) do recur [15] and prognostic markers are needed to select high-risk patients for adjuvant treatment and close monitoring [16]. Traditionally, endometrial cancers are classified into type 1 and type 2. Type 1 endometrial cancers are characterized by being estrogen-related, of endometrioid histology, occurring in pre-menopausal obese women and have more favourable prognosis. Type 2 endometrial cancers are characterised by being non-estrogen related, developing from atrophic endometrium, of non-endometrioid histology, occurring in post-menopausal women and have poorer prognosis [17]. Clinicopathological factors associated with worse survival include older age, higher tumor stage, higher grade, non-endometrioid histology and the presence of lymphovascular space invasion (LVSI) [14].

dMMR is an established carcinogenesis mechanism in endometrial cancer. However, its clinical significance remains controversial. Some studies had showed that dMMR was associated with older age, higher grade, higher stage disease, larger tumor, more LVSI and deeper myometrial invasion [5,18]. By contrast, some studies found no difference in stage, grade or LVSI [16]. dMMR tumors were more often of endometrioid than non-endometrioid histology with dMMR found in 51.4% of endometrioid tumors but only 20% of serous/clear cell tumors [19]. The relationship between dMMR and endometrial cancer survival is still unclear. A recent study involving 728 cases found that in endometrioid endometrial cancer, somatic dMMR was associated with worse disease specific survival (DSS) when compared to pMMR tumors (hazard ratio HR = 2.18) [14]. The worse prognosis was particularly evident in early-stage cancer [3,20] of endometrioid histology [5,14]. A decrease recurrence free survival had also been reported in a small number of studies [5]. Opposing findings were reported in another study involving 473 cases of endometrial cancer, showing that dMMR were associated with improved DSS (HR = 0.3) [6]. No difference in survival was found in some other studies [16]. A meta-analysis failed to find any association between survival and MMR status [21]. 

In additional to prognostic implications, MMR analysis is useful in the development of precision medicine in endometrial cancer [1]. It can guide the delivery of adjuvant therapy [22] and act as a biomarker to predict a response to immunotherapy [23]. Immunotherapy with immune checkpoint inhibitor such as anti-programmed death (PD)-1 and anti-PD-ligand 1 (PD-L1) antibodies is a rapidly emerging anti-cancer therapy [24]. 

The study’s primary objective is to investigate the role of MMR status in the long-term survival of endometrial cancer patients of endometrioid histology. The secondary objectives include determining the prevalence of dMMR in Chinese endometrial cancer patients and its correlation to known prognostic factors such as lymph node involvement, disease grade, disease stage and presence of LVSI. Ninety percent of endometrial cancer is of endometrioid histology and its behavior is very different from non-endometrioid tumors, with the latter having poorer prognosis [16]. In view of the high prevalence of endometrioid tumors, the difference in clinical behavior from non-endometrioid tumors, more dMMR being reported in endometrioid tumors compared to non-endometrioid tumors (51.4% vs. 20%) [3,18,19,25], and the detrimental effect of dMMR being more reported in endometrioid tumors, our study focused on endometrioid tumors.

## 2. Materials and Methods

### 2.1. Patients

This is a retrospective study including 238 cases. Endometrial cancer of endometrioid histology was identified from the electronic database of Prince of Wales Hospital, a Hong Kong public hospital. Patients with hysterectomy performed and histology of surgical specimen analyzed from February 2001 to June 2010 were included. Baseline demographics were collected from the hospital electronic medical record system. Treatment data including operation performed, radiotherapy and chemotherapy were collected. The standard surgical treatments for endometrial cancer include total hysterectomy and bilateral salpingoophorectomy. Pelvic and/or para-aortic lymphadenectomy may be performed depending on the surgical risk, tumor grade, histology, cervical involvement, enlargement of lymph node and depth of myometrial invasion (as assessed by pre-operative endometrial sampling), MRI and intra-operative examination. In general, lymphadenectomy will be performed for cases of stage IB or above or high-grade disease whereas a radical hysterectomy will be performed if cervical invasion is suspected. The need for adjuvant therapy will be discussed in a multi-disciplinary meeting including gynaecologists, clinical oncologists and pathologists.

Pathological data including tumor size (maximum tumor dimension), grade, LVSI, myometrial invasion, cervical invasion, pelvic and para-aortic lymph node involvement, survival data (including 5-year and 10-year disease free survival (DFS), disease specific survival (DSS), overall survival (OS)), and recurrence were collected. Endometrial cancer staging and grading were based on the publication from the International Federation of Gynecology and Obstetrics in 2009. The OS was determined from the date of treatment to date of last contact or death from any cause. The DSS was determined from the date of treatment to date of last contact or death resulting from endometrial cancer. The DFS was determined from the date of treatment to the date of recurrence diagnosis. The study was approved by the local institutional ethics committee (CREC Ref. No. 2019.716).

### 2.2. Mismatch Repair (MMR) Status Analysis with Immunohistochemistry (IHC) 

The tumor expression of MMR proteins including MLH1, MSH2, MSH6 and PMS2 were evaluated by IHC. Immunostaining was performed on 4 μm unstained formalin-fixed paraffin embedded slides with Ventana Optiview detection kit and 3,3′-Diaminobenzidine as chromogen. The list of antibodies used, and protocols adopted are listed in Appendix A. Intensity of immunohistochemical staining was characterized into levels 0, 1, 2, 3 with level ≥1 regarded as a positive stain. MMR protein expression was considered as retained when there was ≥ 10% positive staining in tumor cell nuclei, whereas staining in <10% was considered as indeterminate and 0% staining was considered as loss of expression provided that the internal control was positive [26,27]. If the internal control was negative, the result would be interpreted as non-informative. The tumor was regarded as dMMR if there was a loss of one or more of the four MMR protein expressions [16]. A paired loss of MLH1/PMS2 or MSH2/MSH6 will indicate a defect in the dominant partner (MLH1 or MSH2). The MMR results were interpreted separately by two pathologists.

### 2.3. Statistical Analysis 

Sample size calculation was performed with the online survival curve sample size calculation tool provided by the Centre for Clinical Research and Biostatistics of the Chinese University of Hong Kong [28]. The significance level and power of test were set at 0.05. Median OS was set at 225 months, hazard ratio as 0.42 and rate of dMMR as 20% based on previously published studies [19,29]. The follow-up duration was set at 120 months. A required sample size of 214 was obtained. The data were analysed with software Statistical Package for Social Science Statistics Version 22. Survival was evaluated with Kaplan–Meier survival analysis and compared statistically using a log rank test. Cox regression analysis was used to assess the hazard ratio of MMR status on survival. The association of MMR protein status with clinicopathological parameters was assessed by the Chi square test, Fisher’s exact test and *t*-test. Statistical significance was set at two-sided *p* < 0.05.

## 3. Results

### 3.1. MMR Protein Expression

Protein expression of MLH1, MSH2, MSH6 and PMS2 was evaluated in a total of 238 cases of endometrial cancer of endometrioid histology by IHC. Representative results of immunohistochemical staining of MLH1, MSH2, MSH6 and PMS2 are shown in Figure 1a–e. Cases were considered as MLH1 loss, MSH2 loss, MSH6 loss, and PMS2 loss are shown in Figure 1a, Figure 1b, Figure 1c and Figure 1d, respectively. A representative case with all MMR proteins retained is shown in Figure 1e.

Loss of one or more of the four MMR protein expressions was seen in 43 cases (16.5%), which were regarded as dMMR. Among these cases, 62.8% (27/43) with MLH1 loss, 14% (6/43) with MSH2 loss, 2.3% (1/43) with MSH6 loss, and 20.9% (9/43) with PMS2 loss (Table 1). Therefore, the most common type of deficient MMR protein was MLH1, followed by PMS2. All MMR proteins were retained in 162 cases (62.1%), which were regarded as pMMR. The result was indeterminate in 33 cases (12.6%) (Table 1).

### 3.2. Clinical Characteristics

Two hundred and five cases with deficient MMR (dMMR) or proficient MMR (pMMR) were included in the analysis. The clinical parameters are shown in Table 2. The mean age was larger in the dMMR compared to the pMMR group (57.9 vs. 53.6, *p* = 0.04). Adjuvant pelvic radiotherapy was more frequently given to the dMMR group (*p* = 0.03), while adjuvant vault radiotherapy and chemotherapy were similar. Stage IA endometrial cancer was more common in the pMMR than dMMR group (71% vs. 48.8%, *p* = 0.01), while stage II endometrial cancer was more common in the dMMR than pMMR group (18.6% vs. 8%, *p* = 0.05). The number of early-stage cases (stages I and II) and late-stage cases (stages III and IV) were not different (*p* = 0.18).

### 3.3. Pathological Characteristics

The association of pathological parameters with MMR status is shown in Table 3. Uterine tumors were larger in the dMMR than pMMR group (*p* = 0.01) and dMMR were associated with higher grade disease (*p* = 0.01). Pathological findings in deep myometrial invasion, cervical invasion, LVSI, pelvic and para-aortic lymph node involvement were not different between pMMR and dMMR tumors. Although the difference did not reach statistical significance, there was a trend towards more deep myometrial invasion, cervical invasion and LVSI in the dMMR group (Table 3). 

### 3.4. Survival and Prognosis

The prognostic impact of MMR status was examined in our survival analysis. Throughout the entire cohort, the median follow up was 138 months (range 5 to 223 months). There were 33 deaths (16.1%), median OS was 137 months (range: 5 to 223 months), median DSS was 141 months (range: 14 to 223 months) and median DFS was 137 months (range: 2 to 217 months). For the pMMR group, median follow up was 140 months (range 5 to 223 months). There were 23 deaths (14.2%). For the dMMR group, the median follow up was 122 months (range 17 to 207 months). There were 10 deaths (23.3%), a percentage higher than the pMMR group. The median OS, DSS and DFS are lower than in the dMMR group than the pMMR group, with a median OS of 122 months (range: 17 to 207 months) vs. 139 months (range: 5 to 223 months), median DSS of 123 months (range: 17 to 207 months) vs. 146 months (range: 14 to 223 months) and median DFS of 121 months (range: 2 to 207 months) vs. 139 months (range: 5 to 217 months) respectively. The Kaplan–Meier curves for the OS, DSS and DFS are shown in Figure 2a-c. The rate of 5-year OS/ DSS/ DFS, 10-year OS/ DSS/ DFS and recurrence are shown in Table 4. Ten-year DSS and DFS were significantly higher in the pMMR than dMMR group (94.8% vs. 80%, *p* = 0.02; 87.3% vs. 73.5%, *p* = 0.05) (Table 4). However, the 5-year DSS and DFS only showed a trend of better survival in the pMMR than dMMR group but did not reach statistical significance (Table 4). DFS was longer in the pMMR than dMMR group (*p* = 0.01), with a hazard ratio of 0.25 (95% CI 0.09 to 0.71) (Figure 2b). The DSS was also longer in the pMMR than dMMR group (*p* = 0.01), with a HR of 0.22 (95% CI 0.07 to 0.71) (Figure 2b). However, OS was similar between the two groups (*p* = 0.14) (Figure 2a). The recurrence rate was lower in the pMMR than dMMR group (4.3% vs. 16.3%, *p* = 0.01) (Table 4). 

## 4. Discussion

MMR status has been intensively studied in recent years as a prognostic indicator in endometrial cancer. In colorectal cancer, MSI-H has been shown in multiple studies to be strongly associated with better prognosis [30]; however, results have been contradictory in endometrial cancer. In a meta-analysis including 23 studies [21], no definite evidence between MMR status and detrimental survival in endometrial cancer was found [21]. However, only one out of 23 studies showed an improved overall survival [6] in that meta-analysis and this outlier led to the insignificant conclusion [6]. In that particular study, 20% of cases were of non-endometrioid histology [6]. A study involving 728 cases found a worse DSS in only endometrioid tumors with somatic MMR deficiency (HR 2.18), while no difference was found for germline mutation or when both endometroid and non-endometrioid tumors were included [14]. The detrimental effect of dMMR on endometrial cancer survival was also seen in another study with a HR of 3.25 for DFS and HR of 4.2 for DSS [20]. Other studies only detected poorer survival in early stage endometrioid tumor [3]. These findings reflected the different clinical implication dMMR has on various endometrial cancer subtypes and somatic or germline mutation.

The contradictory results of different studies may be due to lack of stratification of cases into (1) somatic or germline mutation and (2) endometrioid or non-endometrioid cancer. Our study, being the first to focus only on endometrioid tumors, had showed that somatic dMMR in endometrioid endometrial cancer was associated with worse clinicopathological factors including older patient age, larger tumor size, more advanced-stage disease, higher-grade disease and increased need of adjuvant radiotherapy. The poor prognostic implications were reflected in the survival analysis, with more recurrence noticed in the dMMR group and a shorter DFS observed, despite more adjuvant therapy being used in the dMMR group. The poor prognostic implication was particularly profound in the long term, with the 10 years DSS and DFS being significantly higher in the pMMR than dMMR group (94.8% vs. 80%, *p* = 0.02; 87.3% vs. 73.5%, *p* = 0.05) but five-year DSS and DFS only showing a trend of better survival in the pMMR group (96.4% vs. 92.3%, *p* = 0.38; 91.7% vs. 90.7%, *p* = 0.52). The findings of our study further indicated that in a specific group of patients (endometrioid histology), somatic dMMR may be a prognostic indicator.

Early-stage endometrioid histology accounts for >60% of endometrial cancers [31], therefore, our findings are relevant to the majority of endometrial cancer patients. The addition of MMR status in the risk stratification process can potentially identify a proportion of poor prognostic patients in this originally low risk group for individualized treatment and follow up plan. The poor long-term survival in the dMMR group indicated the need of an individualized follow up plan based on risk assessment. Currently, most international guidelines recommend 5-year follow up for endometrial cancer patients [32,33]. However, our results showed that the 5-year survival of dMMR and pMMR group were not very much different and the obvious difference occurs at 5 to 10 years follow up. In the dMMR group, longer follow up to 10 years should be considered. Moreover, more aggressive adjuvant treatment plan may be appropriate for this higher-risk group.

### 4.1. Personalized Medicine in Endometrial Cancer

Traditionally, endometrial cancer had been classified by clinical characteristics such as grade of disease. Recently, there is emerging evidence on the utility of genetic and epigenetic characteristics as prognostic markers for endometrial cancer [34]. The latest 2020 ESGO/ESTRO/ ESP guideline on endometrial cancer recommended encouraging molecular classification in all endometrial cancers [13]. Knowledge of the genetic and epigenetic characteristics of individuals can allow personalized medicine tailored to the individual’s genetic profile. The Cancer Genome Atlas Research Network had classified endometrial cancer into four categories: *POLE* hypermutated, MSI hypermutated, copy number low (*p53* abnormal) and copy number high [35]. Molecular testing is relevant in the risk stratification for prognosis and adjuvant therapy usage, guidance for immunotherapy and pre-screening for Lynch syndrome [13].

### 4.2. Personalized Medicine in Endometrial Cancer—Adjuvant Treatment

Molecular markers are increasingly being incorporated into traditional risk stratification model to identify high risk patients, especially in high-risk endometrial cancers where evidence has shown that molecular biomarkers can serve as better prognostic markers. For example, *POLE* hypermutated high risk endometrial cancer was found to have excellent prognosis, while *p53* abnormal tumours had poor prognosis [13]. Molecular classification can further guide the usage of adjuvant therapy and individualise a follow-up plan based on patients’ molecular characteristics. MMR status has been well demonstrated in colorectal cancer to be an effective predictor for treatment efficacy in adjuvant setting [30], allowing a more tailored adjuvant therapy. In endometrial cancer, there is limited evidence showing a better response rate of dMMR tumors to platinum-based chemotherapy than pMMR tumors (67% vs. 44%) [31]. In a study with 158 endometrial cancer patients receiving chemotherapy and 66 patients receiving radiotherapy, there was a significant increase in OS and PFS in dMMR, non-endometrioid tumors treated with radiotherapy and pMMR stage III/IV patients treated with adjuvant chemotherapy [22]. The majority of endometrial cancers are stage IA disease which is considered as low risk, and adjuvant therapy is not recommended based on current recommendations which utilize clinicopathological factors [36]. With emerging evidence of the adverse effect of dMMR, adjuvant therapy can be considered to improve survival in dMMR early-stage endometrioid endometrial cancer patients. This risk stratification with molecular markers may be a superior model than the existing clinicopathological model or could be an adjunct to formulation of adjuvant treatment. 

### 4.3. Personalized Medicine in Endometrial Cancer—Immunotherapy

Molecular characteristics can predict response to new therapeutic agents such as immunotherapy. For example, MSI analysis has been proven in colorectal cancer to be an effective biomarker to predict response to pembrolizumab, a PD-1 immune checkpoint inhibitors that targets and block PD-1 [37]. The objective response to Pembrolizumab is 40% in the dMMR group compared to 0% in the pMMR group [37]. The evidence with endometrial cancer is not as robust. Nonetheless, pembrolizumab has been approved by the FDA for use in recurrent or metastatic endometrial cancer together with all other types of dMMR tumours [38]. The overall response rate to Pembrolizumab was 39.6%, while the response rate in endometrial cancer alone was 36% [39]. Immunotherapy is very expensive and not without adverse effects. Therefore, the selection of patients who will benefit is important. MMR status can potentially be a biomarker to predict treatment efficacy of PD-1 immune checkpoint inhibitor in endometrial cancer patients [40]. This molecular approach is promising, especially in metastatic recurrent disease where patients would otherwise, be incurable. 

### 4.4. Personalized Medicine in Endometrial Cancer—Lynch Syndrome Screening

Approximately 3% of endometrial cancer patients carry germline MMR gene mutations [13]. A germline mutation in MMR genes results in Lynch syndrome, an inherited cancer syndrome predisposing patients to multiple cancer development, including colorectal and endometrial cancers [41]. Lynch syndrome was found to be present in 5.4% of Chinese endometrial cancer patients [10]. Traditionally, screening of Lynch syndrome has been based on clinical parameters, such as the personal and family history of Lynch-associated malignancies. Somatic loss of MMR expression may be associated with a presence of germline mutation. MSI is a well-established and effective genetic marker for detection of Lynch syndrome [9]. Expanding a universal screening program for Lynch syndrome to include patients with endometrial cancer has been shown to identify 50% more Lynch syndrome patients [42], who would largely be missed by screening based on clinical parameters. The latest guideline from ESGO/ESTRO/ESP and International Society of Gynaecological Pathology recommended testing of MMR with IHC to screen for Lynch syndrome [13]. An early detection can allow appropriate cancer risk reducing intervention to be undertaken in the patients and their relatives, potentially improving their survival.

The strength of our study includes a large sample size of 238 cases focusing on somatic dMMR of endometrioid histology endometrial cancers. The long follow up with a median follow up of more than 10 years is also longer than most of the published studies [3,14,22]. Furthermore, all four MMR protein expressions were evaluated in our study whereas only one to two MMR protein expressions were evaluated in some previous studies [21]. However, the rate of dMMR of 16.5% in our study is less than that reported in other studies [5,6]. This is because we adopted a strict criteria defining dMMR with <1% staining, while staining in <10% was considered as indeterminate, leading to a high proportion of cases being reported as indeterminate (12.6%). If the indeterminate cases were also considered potentially negative, most of the dMMR rates reported in other studies [5,6] would fall into the possible range of dMMR in our cohort (16.5–29.1%). On the other hand, the survival analysis may be affected by the imbalance of age, stage of disease and adjuvant therapy received between the dMMR and pMMR groups. Furthermore, information on other somatic tumour markers such as *p53* and *POLE* status which also affects survival was not available. Lymph node dissection was performed on only 58 cases which is too low to draw any definite conclusion.

## 5. Conclusions

MMR deficiency is present in 16.5% of Hong Kong Chinese women with endometroid endometrial cancers. dMMR is associated with poor clinicopathological factors, worse DFS, DSS and more recurrence. The worse prognosis is particularly evident in the long term at 5 to 10 years follow up.

## Figures and Tables

**Figure 1 biomedicines-09-01385-f001:**
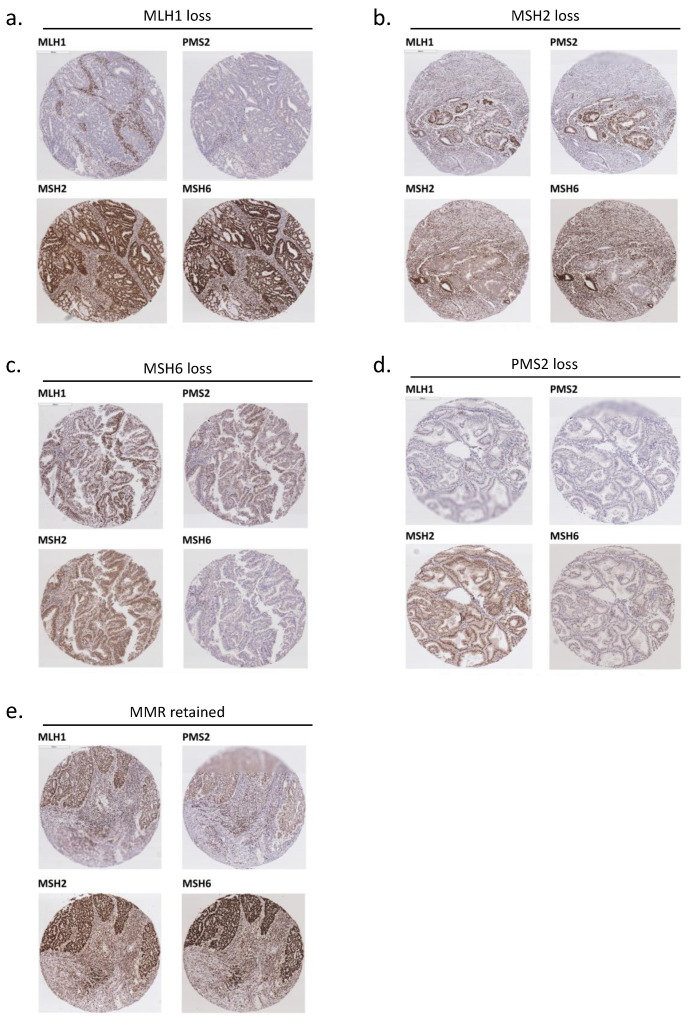
Expression of mismatch repair (MMR) proteins in endometrioid endometrial cancer tissues. (**a**) A representative case of MLH1 loss, immunohistochemistry (IHC) staining showed loss of MLH1 protein expression in tumor cells. (**b**) A representative case of MSH2 loss, IHC staining showed loss of MSH2 protein expression in tumor cells. (**c**) A representative case of MSH6 loss, IHC staining showed loss of MSH6 protein expression in tumor cells. (**d**) A representative case of PMS2 loss, IHC staining showed loss of PMS2 protein expression in tumor cells. (**e**) A representative case of MMR retained, IHC staining showed all four of the MMR proteins (MLH1, MSH2, MSH6 and PMS2) retained in tumor cells.

**Figure 2 biomedicines-09-01385-f002:**
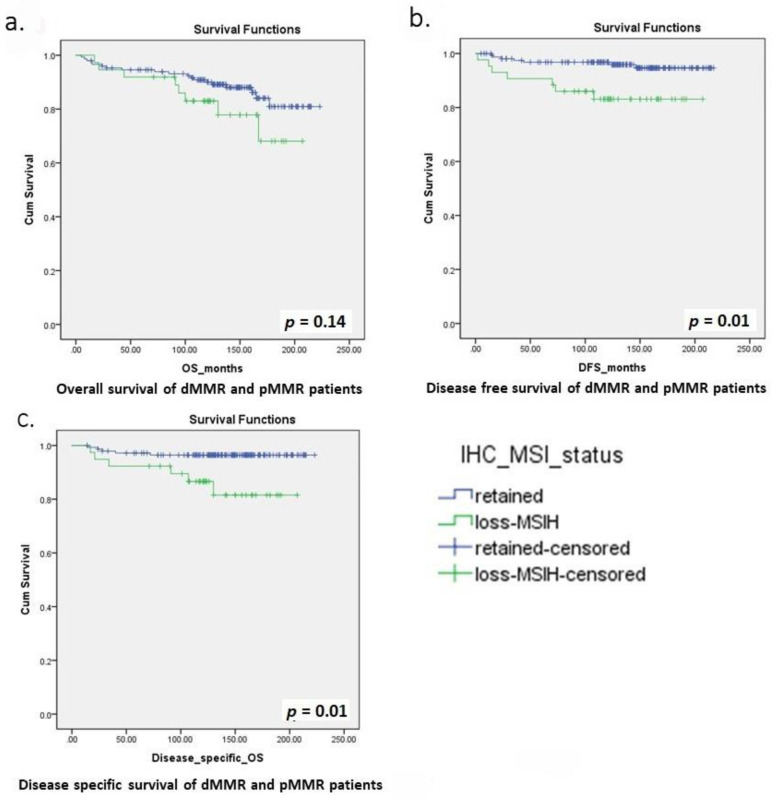
Kaplan–Meier curves for overall survival (OS), disease-free survival (DFS) and disease-specific survival (DSS) of dMMR and pMMR patients. (**a**) overall survival, (**b**) disease-free survival, and (**c**) disease-specific survival of dMMR and pMMR patients.

**Table 1 biomedicines-09-01385-t001:** MMR expression in endometrioid endometrial cancer.

MMR Status	Number [Percentage]
MMR Retained	162/238 [62.1%]
MMR Loss - MLH1 Loss - PMS2 Loss - MSH2 Loss - MSH6 Loss	43/238 [16.5%] - 27/43 [62.8%] - 9/43 [20.9%] - 6/43 [14%] - 1/43 [2.3%]
MMR Indetermined - PMS2 Indeterminate - MSH6 Indeterminate - MLH2 Indeterminate	33/238 [12.6%] - 27/33 [81.8%] - 4/33 [12%] - 2/33 [6%]

**Table 2 biomedicines-09-01385-t002:** MMR status in association with clinical parameters.

Clinical Parameters	MMR Deficient (*n* = 43)	MMR Proficient (*n* = 162)	*p*-Value
Age	Mean 57.9 (SD 11.29)	Mean 53.6 (SD 12.29)	0.04
Parity *(missing n = 62)*	Mean 2.33 (SD 2.19)	Mean 1.82 (SD 1.49)	0.14
Parity -0-1-2-3-≥4	*(missing n = 13)* -9/30 ((30%)-2/30 (6.75)-7/30 (23.3%)-5/30 (16.7%)-7/30 (23.3%)	*(missing n = 49)* -30/113 (26.5%)-17/113 (15%)-30/113 (26.5%)-22/113 (19.5%)-14/113 (12.4%)	
Menopause	22/40 (55%)	76/152 (50%)	0.7
Colorectal cancer	3/43 (7%)	4/162 (2.5%)	0.16
Operation -Total hysterectomy-Radical Hysterectomy-Pelvic lymphadenectomy-Para-aortic lymphadenectomy	-38/43 (88.4%)-5/43 (11.6%)-12/43 (27.9%)-10/43 (23.3%)	-156/162 (96.3%)-6/162 (3.7%)-46/162 (28.4%)-26/162 (16%)	0.08
Bilateral salpingoophorectomy	43/43 (100%)	153/162 (94.4%)	0.25
Adjuvant vault radiotherapy	10/43 (23.3%)	24/162 (14.8%)	0.28
Adjuvant pelvic radiotherapy	18/43 (41.9%)	38/162 (23.5%)	0.03
Adjuvant chemotherapy	3/43 (7%)	12/162 (7.4%)	0.61
Stage -IA-IB-II-IIIA-IIIB-IIIC1-IIIC2	-21/43 (48.8%)-8/43 (18.6%)-8/43 (18.6%)-2/43 (4.7%)-1/43 (2.3%)-1/43 (2.3%)-2/43 (4.7%)	-115/162 (71%)-21/162 (13%)-13/162 (8%)-7/162 (4.3%)-2/162 (1.2%)-1/162 (0.6%)-3/162 (1.9%)	
0.01

0.05





**Table 3 biomedicines-09-01385-t003:** MMR status associated with pathological parameters.

Pathological Parameters	MMR Deficient (*n* = 43)	MMR Proficient (*n* = 162)	*p*-Value
Uterine tumour size (cm)	Mean 3.39 (SD 2.26)	Mean 2.46 (SD 1.89)	0.01
Grade - 1 - 2 - 3	- 24/43 (55.8%) - 14/43 (32.6%) - 5/43 (11.6%)	- 131/162 (80.9%) - 26/162 (16%) - 5/162 (3.1%)	0.01
Deep myometrial invasion	13/43 (30.2%)	28/162 (17.3%)	0.09
Cervical invasion	11/43 (25.6%)	20/162 (12.3%)	0.06
Lymphovascular space invasion (LVSI) (missing *n* = 88)	10/26 (38.5%)	17/91 (18.7%)	0.07
Pelvic lymph node involved	2/12 (16.7%) *(Not performed n = 31)*	3/46 (6.5%) *(Not performed n = 116)*	0.59
Para-aortic lymph node involved	2/10 (20%) *(Not performed n = 33)*	3/26 (11.5%) *(Not performed n =136)*	0.43

**Table 4 biomedicines-09-01385-t004:** MMR status in association with survival parameters.

Survival Parameters	MMR Deficient (*n* = 43)	MMR Proficient (*n* = 162)	*p*-Value
5 years overall survival	39/43 (90.7%) (Loss to follow up n = 0)	145/157 (92.4%) (Loss to follow up n = 5)	0.46
10 years overall survival	26/34 (76.5%) (Loss to follow up n = 9)	117/134 (87.3%) (Loss to follow up n = 28)	0.19
5 years disease-specific overall survival	36/39 (92.3%) (Loss to follow up n = 0)	134/139 (96.4%) (Loss to follow up n = 5)	0.38
10 years disease-specific overall survival	24/30 (80%) (Loss to follow up n = 9)	110/116 (94.8%) (Loss to follow up n = 28)	0.02
5 years disease-free survival	39/43 (90.7%) (Loss to follow up n = 0)	144/157 (91.7%) (Loss to follow up n = 5)	0.52
10 years disease-free survival	25/34 (73.5%) (Loss to follow up n = 9)	117/134 (87.3%) (Loss to follow up n = 28)	0.05
Disease recurrence	7/43 (16.3%)	1/162 (4.3%)	0.01

## Data Availability

The data presented in this study are available on request from the corresponding author. The data are not publicly available due to privacy.

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
