# Peer review of "Long-Term Survival and Clinicopathological Implications of DNA Mismatch Repair Status in Endometrioid Endometrial Cancers in Hong Kong Chinese Women"

_biomedicines, 2021, doi:10.3390/biomedicines9101385_

Round 1
Reviewer 1 Report
The authors of the study aim to determine the importance of the defects in the components of the mismatch DNA repair machinery dMMR as a predictive factor in disease progression of endometrioid carcinoma. The study is based on a retrospective analysis of 238 endometrial cancer cases- the cohort of Chinese Women in Hongkong. Authors classify as dMMR all cases where one of the 4 markers is reduced (MLH1, MSH2, MSH6, and PMS2) and present convincing and high-quality images to illustrate the findings. The data set has been analyzed independently with two independent pathologists which important control mechanism. Follow-up statistical analysis and interpretation of the data do show a significant reduction in long-term 10-year survival for dMMR patients, but this only mild difference after 5 years. Particular caution in data interpretation needs to be exerted because the dMMR group was significantly older which authors acknowledge briefly in the discussion.
While the methodology of the study and systematic approach as well as reasonably large sample size, do warrant publication as it could help experts in the field to complement at the moment still incomplete picture of the role of dMMR in endometrial cancer, some revision to the manuscript is necessary.
This is especially the case for the interpretation of the data and some claims which are not supported by the experimental data. For example, the authors make a very firm statement equalizing in general IHC methodology with sequencing and PCR data ( based on literature) and go on discussing presented staining as proof that cases in question do have somatic mutations. However, the study does not entail either data about germline mutations of the patients or shows mutational profiles of the tumors. This needs to be clarified and statements toned down. Also, the authors group all different defects together, although they have two relatively large subgroups MLH1 ( 62%) and PMS2 (21%). Did they perform any subgroup analysis, as this would be an interesting aspect?
Overall this is an interesting data set, and could be an interesting topic for the readership of “Biomedicines”.
Author Response
Reviewer 1
- This is especially the case for the interpretation of the data and some claims which are not supported by the experimental data. For example, the authors make a very firm statement equalizing in general IHC methodology with sequencing and PCR data (based on literature) and go on discussing presented staining as proof that cases in question do have somatic mutations. However, the study does not entail either data about germline mutations of the patients or shows mutational profiles of the tumors. This needs to be clarified and statements toned down.
Response: Thank you for your comment. Another study which showed a lower accuracy of IHC in detecting dMMR when compared to PCR had been added to tone down the statement (revised on page 2 line 67-72).
Reviewer 2 Report
The paper is well written, the research is original and the methodology well structured. A paragraph should be added in the discussion to further emphasize the importance of personalizing the care of patients with endometrial cancer and add the following references:
- Cavaliere AF, Perelli F, Zaami S, Piergentili R, Mattei A, Vizzielli G, Scambia G, Straface G, Restaino S, Signore F. Towards Personalized Medicine: Non-Coding RNAs and Endometrial Cancer. Healthcare (Basel). 2021 Jul 30;9(8):965. doi: 10.3390/healthcare9080965. PMID: 34442102; PMCID: PMC8393611.
- Piergentili R, Zaami S, Cavaliere AF, Signore F, Scambia G, Mattei A, Marinelli E, Gulia C, Perelli F. Non-Coding RNAs as Prognostic Markers for Endometrial Cancer. Int J Mol Sci. 2021 Mar 19;22(6):3151. doi: 10.3390/ijms22063151. PMID: 33808791; PMCID: PMC8003471.
- Concin N, Matias-Guiu X, Vergote I, Cibula D, Mirza MR, Marnitz S, Ledermann J, Bosse T, Chargari C, Fagotti A, Fotopoulou C, Martin AG, Lax S, Lorusso D, Marth C, Morice P, Nout RA, O'Donnell D, Querleu D, Raspollini MR, Sehouli J, Sturdza A, Taylor A, Westermann A, Wimberger P, Colombo N, Planchamp F, Creutzberg CL. ESGO/ESTRO/ESP guidelines for the management of patients with endometrial carcinoma. Radiother Oncol. 2021 Jan;154:327-353. doi: 10.1016/j.radonc.2020.11.018. PMID: 33712263.
Author Response
Reviewer 2
The paper is well written, the research is original and the methodology well structured. A paragraph should be added in the discussion to further emphasize the importance of personalizing the care of patients with endometrial cancer and add the following references:
- Cavaliere AF, Perelli F, Zaami S, Piergentili R, Mattei A, Vizzielli G, Scambia G, Straface G, Restaino S, Signore F. Towards Personalized Medicine: Non-Coding RNAs and Endometrial Cancer. Healthcare (Basel). 2021 Jul 30;9(8):965. doi: 10.3390/healthcare9080965. PMID: 34442102; PMCID: PMC8393611.
- Piergentili R, Zaami S, Cavaliere AF, Signore F, Scambia G, Mattei A, Marinelli E, Gulia C, Perelli F. Non-Coding RNAs as Prognostic Markers for Endometrial Cancer. Int J Mol Sci. 2021 Mar 19;22(6):3151. doi: 10.3390/ijms22063151. PMID: 33808791; PMCID: PMC8003471.
- Concin N, Matias-Guiu X, Vergote I, Cibula D, Mirza MR, Marnitz S, Ledermann J, Bosse T, Chargari C, Fagotti A, Fotopoulou C, Martin AG, Lax S, Lorusso D, Marth C, Morice P, Nout RA, O'Donnell D, Querleu D, Raspollini MR, Sehouli J, Sturdza A, Taylor A, Westermann A, Wimberger P, Colombo N, Planchamp F, Creutzberg CL. ESGO/ESTRO/ESP guidelines for the management of patients with endometrial carcinoma. Radiother Oncol. 2021 Jan;154:327-353. doi: 10.1016/j.radonc.2020.11.018. PMID: 33712263.
Response: Thank you for your comment. Paragraphs had been added to further emphasize the importance of personalizing the care of patients with endometrial cancer with regards to 1) Adjuvant therapy, 2) Immunotherapy, and 3) Lynch Syndrome screening. The three references had been added to the discussion (revised from page 11 line 311 to page 12 line 378).